# The Citrus Flavonoid Hesperetin Encounters Diabetes-Mediated Alzheimer-Type Neuropathologic Changes through Relieving Advanced Glycation End-Products Inducing Endoplasmic Reticulum Stress

**DOI:** 10.3390/nu14040745

**Published:** 2022-02-10

**Authors:** Mei-Chou Lai, Wayne-Young Liu, Shorong-Shii Liou, I-Min Liu

**Affiliations:** 1Department of Pharmacy and Master Program, Collage of Pharmacy and Health Care, Tajen University, Pingtung County 90741, Taiwan; meei@tajen.edu.tw (M.-C.L.); ssliou@tajen.edu.tw (S.-S.L.); 2Department of Urology, Jen-Ai Hospital, Taichung City 41265, Taiwan; waynedoctor@gmail.com; 3Center for Basic Medical Science, Collage of Health Science, Central Taiwan University of Science and Technology, Taichung City 40601, Taiwan

**Keywords:** hesperetin, advanced glycation end-products, Alzheimer’s disease, SH-SY5Y cells, endoplasmic reticulum stress

## Abstract

The present study investigates whether hesperetin, a citrus flavonoid, can encounter advanced glycation end-product (AGE)-induced Alzheimer’s disease-like pathophysiological changes with the underlying mechanisms. SH-SY5Y cells pretreated with hesperetin before stimulation with AGEs (200 μg/mL) were assessed in the following experiments. Hesperetin (40 μmol/L) elevated the reduced cell viability induced by AGEs. Hesperetin ameliorated reactive oxygen species overproduction and the downregulation of superoxide dismutase, glutathione peroxidase, and catalase, triggered by AGEs. Amyloid precursor protein upregulation, accompanied by the increased production of Aβ, caused by AGEs, was reversed by hesperetin. However, hesperetin lowered β-site APP-cleaving enzyme 1 expression, inducing insulin-degrading and neprilysin expression. In addition, hesperetin downregulated the expressions of the AGEs-induced endoplasmic reticulum (ER) stress proteins, including 78-kDa glucose-regulated protein and C/EBP homologous protein, and lowered the phosphorylation of protein kinase R-like ER kinase and activating transcription factor 4. Hesperetin-pretreated cells had a minor apoptotic DNA fragmentation. Hesperetin is able to upregulate Bcl-2 protein expression, downregulate Bax expression, and decrease caspase-12/-9/-3 activity as well, indicating that it inhibits ER stress-mediated neuronal apoptosis. There is a similar effect between hesperetin and positive rosiglitazone control against Aβ aggravation of SH-SY5Y cell injury induced by AGEs. Thus, hesperetin might be a potential agent for treating glycation-induced Aβ neurotoxicity.

## 1. Introduction

Alzheimer’s disease (AD) is one of the neurodegenerative diseases characterized by progressive memory deficits and cognitive decline [1]. The neuropathological symptoms of AD feature amyloid plaques, neurofibrillary tangles with neurons, and synapses loss in the brain areas linked to cognition, learning, and memory [2]. Based on the amyloid cascade hypothesis, the accumulation of amyloid β (Aβ) is a significant cause of AD [3]. Although available therapeutic methods deliver moderate relief of cognitive symptoms that cannot significantly alter these disease’s progression due to AD’s multifactorial and complex nature, successful therapeutic and preventive interventions are strongly needed [4].

An increasing epidemiological display shows a higher risk of developing AD in diabetic patients [5]. The pathology associated with diabetes-mediated AD involves chronic hyperglycemia accelerating the reaction between glucose and proteins that eventually cause advanced glycation end-product (AGE) formation [6]. AGEs bind to the cell surface receptors (RAGEs) to trigger reactive oxygen species (ROS) production, leading to downstream activation pathways associated with amyloid precursor protein (APP) processing and Aβ production [7]. The clearance of deficiencies of Aβ participates in the accumulation and aggregation of Aβ, which interfere with several cellular processes and result in endoplasmic reticulum (ER) stress, to trigger a complex multistep cascade, which finally causes neuronal death and dementia [8]. Once under ER stress, the 78-kDa glucose-regulated protein (GRP78) sensor can activate the unfolded protein response (UPR) branches to restore ER homeostasis [9]. However, UPR fails to fix the normal function of ER during prolonged or overwhelming ER stress, and can trigger pro-apoptotic signals by activating the transcription factor C/EBP homologous protein (CHOP) [10]. CHOP represses the B-cell lymphoma 2 (Bcl-2) gene expression, thus downregulating the anti-apoptotic Bcl-2 protein, but promoting the activation and translocation of pro-apoptotic proteins, such as the Bcl-2-associated X protein (Bax), to mitochondria [10]. In addition to mediating by a transcription factor-dependent pathway, the UPR can elicit apoptosis by another mechanism: a caspase-dependent path [11]. Caspase-12 is vital in conducting the caspase-dependent UPR. Upon UPR activation, caspase-12 translocates from the ER to cleave and stimulate effector caspase-9 and caspase-3 [11].

As mentioned above, disrupting the AGE/RAGE signaling cascade and blockage of ER stress-mediated neuronal apoptosis may be a potential target for Aβ-mediated aggravation of AD pathology. Rosiglitazone, a thiazolidinedione, is therapeutically used to combat hyperglycemia in metabolic syndrome and type 2 diabetes [12]. Evidence confirms that rosiglitazone protects the cell against oxidative stress, inflammation, and apoptosis, and lessens AGE-induced damage in neural stem cells [13]. In *in vivo* models of AD, rosiglitazone significantly improved memory and cognition deficits; these beneficial effects were observed in AD patients as well [14,15]. Thus, anti-diabetic drugs could be positive in treating AD.

Phytonutrients, also known as phytochemicals from plants, exhibit different biological activities due to the regulation of several cellular molecular pathways and are believed to have practical value for human health [16]. Among the phytonutrients, flavanones have the highest bioavailability of flavonoid compounds [17]. Flavonoids play significant roles in neuroprotective applications, such as protecting against the development of dementia [18], promoting neuronal survival in nutrient deprivation conditions [19], and the emerging role as therapeutics against AD [20]. Hesperetin, a flavanone derivative, is widely distributed among plants and found in daily diets, predominantly in fruits and vegetables, which exhibits many beneficial actions, such as lessening the deterioration of diseases, including diabetes, hypertension, and arteriosclerosis [21]. As oxidative stress and inflammation are closely related to the onset/progression of AD, it revealed that the beneficial effects of hesperetin in neurodegeneration are highly dependent on the antioxidant and anti-inflammatory activities [22]; however, the protective mechanisms of hesperetin would require an in-depth investigation.

The severity of Aβ accumulation as proportional to oxidative stress in SH-SY5Y cells was observed [23]. Thus, it is common to use the retinoic acid-differentiated SH-SY5Y cells line to study the effects of natural compounds against the AGE-induced AD-like pathophysiological changes [24]. Our study used rosiglitazone as a positive drug to determine whether hesperetin has a protective impact on AGEs associated with Aβ neurotoxicity in SH-SY5Y cells, and to explore the possible mechanisms.

## 2. Materials and Methods

### 2.1. Cell Culture

Human SH-SY5Y neuroblastoma cells (no. CRL-266) obtained from the American Type Culture Collection (Manassas, VA, USA) were used for the experiments. The cells were cultured in DMEM/F12 medium (Sigma-Aldrich, St. Louis, MO, USA) supplemented with 10% fetal bovine serum, 1% nonessential amino acids (Gibco, Waltham, MA, USA), 100 IU/mL penicillin, and 100 mg/mL streptomycin (Sigma-Aldrich, St. Louis, MO, USA) at 37 °C in a humidified 5% CO_2_ incubator. SH-SY5Y cells were seeded at 1 × 10^6^ cells/cm^2^ in 100 mm diameter culture dishes in DMEM/F12 medium containing 10% FBS to induce neuronal differentiation. Once the cells were 40–50% confluent, differentiation was initiated by adding 10 mmol/L of retinoic acid (Sigma-Aldrich, St. Louis, MO, USA). Treatments for all the experiments were carried out after the five-day differentiation period.

### 2.2. AGE Induction and Treatments

Cells were seeded at a density of 2 × 10^6^ cells per well in 6-well plates. Cultures were dissociated in 0.05% (*w*/*v*) trypsin (Sigma-Aldrich, St. Louis, MO, USA) in a phosphate-buffered saline (PBS), pH 7.4, and then passaged when the cell was confluent. For AGE-induction studies, cells were cultured in a fresh medium with 1% FBS for 2 h. Later, cells were pretreated with hesperetin (Sigma-Aldrich, St. Louis, MO, USA; Cat. # 69097-99-0, purity ≥ 95%) at different concentrations (10–40 μmol/L), or rosiglitazone (Sigma-Aldrich, St. Louis, MO, USA; Cat. # 557366, purity, ≥99%) at 10 μmol/L [25] for 1 h, followed by exposure to appropriate concentrations of AGEs in bovine serum albumin (AGEs-BSA) (Sigma-Aldrich, St. Louis, MO, USA; 50–250 μg/mL) for different time points (6–48 h) without medium change. The stock solution (100 μmol/L) of hesperetin or rosiglitazone was made by dimethyl sulfoxide (DMSO, Sigma-Aldrich, St. Louis, MO, USA) and diluted in culture medium to the appropriate concentrations for subsequent experiments. The final concentration of DMSO was ≤0.1%, a level generally harmless to most cells [26]. The following experiments were assessed after treatment.

### 2.3. Cell Viability Assay

The 3-(4,5-dimethyl thiazol-2-yl)-2,5-diphenyl tetrazolium bromide (MTT; Sigma-Aldrich, St. Louis, MO, USA) assay used to measure the cellular metabolic activity could be an indicator of cell viability [27]. Cells were exposed to the different concentrations of hesperetin (10–40 μmol/L) or rosiglitazone (10 μmol/L) in the medium for 24 h at 37 °C in 5% CO_2_ in the air. MTT (5 mg/mL of stock in PBS) was added (20 μL/well in 200 μL of cell suspension), then the medium was aspirated and the cells were incubated with MTT dye for another 4 h. At the end of incubation, the dye was taken out and 100 μL of DMSO was added to each well to solubilize the MTT formazan crystals. The absorbance of the solution in each well was measured on a microplate reader (SpectraMax M5, Molecular Devices, Sunnyvale, CA, USA) at 540 nm, and background optical densities were subtracted from that of each well. The net absorbance from the wells of the cells cultured with the vehicle (not treated) was taken as the 100% viability. The data are expressed as percentages compared to the vehicle-treated control.

### 2.4. Determination of Intracellular ROS Levels and Antioxidant Enzyme Activities

Cells were reacted with 10 μmol/L dichloro-dihydro-fluorescein diacetate (Sigma-Aldrich, St. Louis, MO, USA) at 37 °C in the dark for 10 min. The change in dichlorofluorescein fluorescence intensity was measured at an excitation wavelength of 488 nm and an emission wavelength of 530 nm on a microplate reader (SpectraMax M5, Molecular Devices, Sunnyvale, CA, USA).

The determination of antioxidant enzyme activity, including superoxide dismutase (SOD; EC 1.15.1.1), glutathione peroxidase (GSH-Px; EC 1.11.1.9), and catalase (CAT; EC 1.11.1.6), were performed using an SOD Activity Colorimetric Assay Kit (Cat. # K335), GPx Colorimetric Assay Kit (Cat. # K762), and CAT Activity Colorimetric Assay Kit (Cat. # K773), respectively, following the procedures provided by Bio Vision, Inc. (San Francisco, CA, USA). The assay for SOD activity was determined by inhibiting the nitroblue tetrazolium reduction, with a xanthine-xanthine oxidase used as a superoxide generator [28]. The production of formazan was determined spectrophotometrically at 450 nm. The activity of GSH-Px was estimated based on the assay after the reduction in absorbance at 340 nm due to reduced nicotinamide adenine dinucleotide phosphate being consumed by glutathione reductase, with the incubation of the sample in the presence of hydrogen peroxide (H_2_O_2_) to starting the reaction [29]. The activity of CAT was determined from the decomposition rate of H_2_O_2_ to water and oxygen [30], and the absorbance was measured at 570 nm in a microplate reader (SpectraMax M5, Molecular Devices, Sunnyvale, CA, USA). Enzyme activities were expressed as units per milligram protein. The protein concentration was measured by using a Bio-Rad protein assay.

### 2.5. Measurements of Aβ_1–40_ and Aβ_1–42_

The specific enzyme-linked immunosorbent assay (ELISA) kits, Amyloid β (1–40) Human Assay Kit (Cat. # KHB348, Invitrogen, California, USA) and Amyloid β (1–42) Human Assay Kit (Cat. # KHB3441, Invitrogen, California, USA), were used to quantify Aβ_1–40_ and Aβ_1–42_ in cells. The optical density at 450 nm was measured using a microplate reader (Spectramax M5, Molecular Devices, Sunnyvale, CA, USA).

### 2.6. Measurements of Caspase-12, -9, and -3 Activities

A fluorometric substrate kit (Abcam plc., Cambridge, U.K., Cat. # ab65664), based on detecting the cleavage of the substrate ATAD-AFC (ATAD: acetyl-alanine-threonine-alanine-aspartic acid; AFC: 7-amino-4-trifluoromethyl coumarin), was used to measure caspase-12-like activity. In brief, aliquots of cellular extracts containing 300 μg of protein were reacted with the ATAD-AFC substrate (final concentration 50 μmol/L) in a reaction buffer and incubated at 37 °C for 2 h. The fluorescent detection of free AFC after being cleaved from the peptide substrate, ATAD-AFC, at Ex/Em = 400/505 nm was quantified as the caspase-12-like activity.

The activities of caspase-9 (Abcam plc., Cambridge, U.K., Cat. # ab65608) and -3 (Abcam plc., Cambridge, U.K., Cat. # ab39401) were measured by the colorimetric assay kits. In brief, aliquots of cell extracts containing 50 or 100 μg of protein were incubated for 2 h at 37 °C in a reaction buffer, with 100 μmol/L chromogenic substrates for caspase-9 (Ac-LEHD-pNA), or caspase-3 (Ac-DEVD-pNA). The caspase-9/-3-like activity was determined by measuring the cleavage substrate at 405 nm using a microplate reader. The results were expressed as relative to those obtained with vehicle-treated control.

### 2.7. Measurement of DNA Fragmentation

The determination of the cytoplasmic histone-associated DNA fragments (mono- and oligo nucleosomes) after induced cell death was quantitated by a cellular DNA fragmentation ELISA kit (Roche Molecular Biochemicals, Mannheim, Germany) with a primary anti-histone mouse monoclonal antibody coated to the microtiter plate, and a second anti-DNA mouse monoclonal antibody coupled to peroxidase [31]. The retained peroxidase in the immunocomplex was determined by incubating 2,2′-azino-di-[3-ethylbenzthiazoline sulfonate] as a substrate for 10 min at 20 °C. The color change was measured at a wavelength of 405 nm using a microplate reader (Spectramax M5, Molecular Devices, Sunnyvale, CA, USA). The optical density values at 405 nm were corrected for the total protein present in each sample. The data were normalized to the vehicle-treated control.

### 2.8. Western Blot Analysis

Cells were lysed with a lysis buffer containing a mixture of 30 mmol/L Tris-HCl, pH 7.4, 250 mmol/L Na_3_VO_4_, 5 mmol/L EDTA, 250 mmol/L sucrose, 1% Triton X-100 with protease inhibitor, and a phosphatase inhibitor cocktail. The Bio-Rad protein assay kit was used to determine the protein concentrations. The proteins (50 μg/lane) were loaded on a 12.5% polyacrylamide gel, submitted to electrophoresis, and then electroblotted onto polyvinylidene fluoride membranes (Millipore, Bedford, MA, USA). For the renaturation of proteins and the blocking of the unoccupied spaces, the membrane was incubated overnight at 4 °C in 0.1% Tween 20-PBS containing 5% nonfat dry milk.

After blocking, the membranes were then incubated overnight at 4 °C with the following primary antibodies: APP (Cell Signaling Technology, Beverly, CA, USA, Cat. # 2452); β-site APP-cleaving enzyme (BACE)1 (Santa Cruz Biotechnology, Inc., Santa Cruz, CA, USA, Cat. # sc-33711); insulin degrading enzyme (IDE; Santa Cruz Biotechnology, Inc., Santa Cruz, CA. USA, Cat. # sc-393887); neprilysin (NEP; Santa Cruz Biotechnology, Inc., Santa Cruz, CA. USA, Cat. # sc-9149); GRP78 (Santa Cruz Biotechnology, Inc., Santa Cruz, CA. USA, Cat. # sc-13539); protein kinase-like endoplasmic reticulum kinase (PERK; Cell Signaling Technology, Beverly, CA, USA, Cat. # 5683); p-PERK (Thr980) (Cell Signaling Technology, Beverly, CA, USA, Cat. # 3179); eukaryotic initiation factor (eIF) 2α (Cell Signaling Technology, Beverly, CA, USA, Cat. # 5324); p-eIF2a (Ser51) (Cell Signaling Technology, Beverly, CA, USA, Cat. # 9721); activating transcription factor (ATF) 4 (Cell Signaling Technology, Beverly, CA, USA, Cat. # 11815); CHOP (Cell Signaling Technology, Beverly, CA, USA, Cat. # 2895); Bcl-2 (Cell Signaling Technology, Beverly, CA, USA, Cat. # 3948); and Bax (Cell Signaling Technology, Beverly, CA, USA Cat. # 14796). The β-actin antibody (Santa Cruz Biotechnology, Inc., Santa Cruz, CA, USA, Cat. #sc-8432) was used as an internal control in immunoblotting. All the antibodies were utilized at 1:1000 dilution. The membranes were washed 3 times with 0.1% *v*/*v* Tween 20 in PBS and subsequently incubated for 1 h at room temperature with a horseradish peroxidase secondary antibody (1:4000). All immunoblots were visualized by enhanced chemiluminescence (Amersham Biosciences,. Buckinghamshire, UK). The immunoreactive protein in each band was quantitatively determined by a densitometer using the ATTO Densitograph Software (ATTO Corp., Tokyo, Japan). The densitometry units for the ontrol samples (not treated) on each immunoblot were regarded as 1.0. The values of the experimental samples were presented relatively to this adjusted mean value.

### 2.9. Statistical Aanalysis

The results are presented as the mean ± standard deviation (SD) of five independent experiments performed triplicate (n = 5). The evaluation of statistically significant differences, one-way analysis of variance, and Dunnett range post hoc comparisons using Systat SigmaPlot version 14.0 (Systat Software Inc., San Jose, CA, USA) was used. It was considered statistically significant when the differences were at *p* < 0.05.

## 3. Results

### 3.1. Hesperetin Improves Cell Viability in AGE-Challenged Cells

The cell viability was reduced from 88.4 to 57.1%, respectively, under the condition when SH-SY5Y cells were incubated with AGEs at 50 to 250 μg/mL concentrations for 24 h (Figure 1A). The AGE (200 μg/mL)-induced cell viability reduction presented a time-dependent tendency in SH-SY5Y cells (Figure 1B). In this study, SH-SY5Y cells were exposed for 24 h to 200 μg/mL to start neuronal insults in the subsequent experiments. The 1 h pretreatment of hesperetin before the exposure of AGEs (200 μg/mL) improved the cell viability associated with an increasing concentration of hesperetin (10–40 μmol/L; Figure 1C). The protective potency of hesperetin was most potent at the 40 μmol/L concentration. The survival rate sustained to 83.9% in SH-SY5Y cells treated with hesperetin (40 μmol/L) before the AGE’s (200 μg/mL) exposure (Figure 1C). Pretreatment at 10 μmol/L of rosiglitazone with AGE (200 μg/mL)-cultured cells obtained a similar result in cell viability (84.3%; Figure 1C). Neither hesperetin nor rosiglitazone alone could induce cell death (Figure 1D).

### 3.2. Hesperetin Alleviates Oxidative Stress in Cells under AGE Stimulation

The AGEs elevated the intracellular ROS production to 2.5-fold in untreated controls, which was reduced by hesperetin in a concentration-dependent manner (Figure 2A). Hesperetin (40 μmol/L) pretreatment reduced the AGE-induced ROS production in SH-SY5Y cells by 35.9%, relative to the values from vehicle-treated counterparts; the results were similar to the effects produced by 10 μmol/L rosiglitazone (Figure 2A).

The activities in SOD and GSH-Px were reduced markedly when SH-SY5Y cells were exposed to 24 h AGEs (200 μg/mL), but were abrogated by hesperetin pretreatment in a concentration-dependent manner (Figure 2B,C). The SOD and GSH-Px activity reductions in AGE-exposed SH-SY5Y cells were reversed by rosiglitazone (10 μmol/L), with results similar to those produced by hesperetin (40 μmol/L; Figure 2B,C).

The CAT activity in SH-SY5Y cells under AGE stimulation was lower (25.9%) than the untreated control group (Figure 2D). The pretreatment of SH-SY5Y cells with hesperetin (40 μmol/L) or rosiglitazone (10 μmol/L) increased the CAT activity to 3.1- and 3.3-fold, respectively, in AGEs incubated relative to those observed in the vehicle-treated counterparts (Figure 2D).

### 3.3. Hesperetin Lessens APP Cleavage and Decreases Aβ Secretion in Cells under AGE Stimulation

AGEs elevated the APP protein levels in SH-SY5Y cells to 4.8-fold of that in the vehicle-treated group (Figure 3A). The AGE-induced upregulation of APP expression was reduced in SH-SY5Y cells pretreated with hesperetin (40 μmol/L) or rosiglitazone (10 μmol/L) by 55.3 and 58.8%, respectively, relative to those in the vehicle-treated counterpart group (Figure 3A).

The BACE1 level in SH-SY5Y cells receiving AGE induction was 4.4-fold of the vehicle-treated group. Hesperetin (40 μmol/L) pretreatment reduced the higher BACE1 levels by 51.1% (Figure 3A). In contrast, rosiglitazone did not influence BACE1 levels in SH-SY5Y cells that only received AGE induction (Figure 3A).

The protein levels of IDE and NEP in AGE-cultured SH-SY5Y cells were lowered to 43.5 and 45.6% of those in the untreated controls, respectively. At the same time, hesperetin (40 μmol/L) or rosiglitazone (10 μmol/L) pretreatment resulted in the upregulation of both protein levels of IDE and NEP (Figure 3A).

The results show that AGEs caused 3.3-fold and 2.6-fold increases in the amount of Aβ1-40 and Aβ1-42 in SH-SY5Y cells, respectively, whereas rosiglitazone (10 μmol/L) pretreatment attenuated these enhancements (Figure 3B). It decreased to 50.7 and 54.1% in Aβ1-40 and Aβ1-42 levels, respectively, when SH-SY5Y cells received hesperetin (40 μmol/L) treatment before AGE stimulation (Figure 3B).

### 3.4. Hesperetin Attenuated ER Stress in Cells under AGE Stimulation

AGEs increased the GRP78 levels in SH-SY5Y cells to 4.5-fold of the vehicle-treated control (Figure 4A). Hesperetin (40 μmol/L) or rosiglitazone (10 μmol/L) lowered the AGE’s induced higher GRP78 levels in SH-SY5Y cells by 44.9 and 52.6%, respectively (Figure 4A).

The ratio of phosphoprotein to total protein in PERK (p-PERK/PERK) and eIF2α (p-eIF2α/eIF2α) are 4.1- and 4.3-fold more significant in SH-SY5Y cells under AGE induction (Figure 4B,C). Hesperetin (40 μmol/L) downregulates the AGEs that induced the increase in p-PERK/PERK and p-eIF2α/eIF2α to 2.2- and 2.1-fold (Figure 4B,C). The elevation ratios of p-PERK/PERK and p-eIF2α/eIF2α induced by AGEs were decreased by rosiglitazone (10 μmol/L) to 1.9- and 2.0-fold, respectively (Figure 4B,C). Hesperetin and rosiglitazone made no changes to the total protein of PERK and eIF2α (Figure 4B,C).

AGEs caused the protein levels of ATF4 and CHOP to increase by 4.8- and 4.0-fold in SH-SY5Y cells, respectively (Figure 4D,E). AGE-induced upregulation on ATF4 and CHOP protein expression in SH-SY5Y cells remarkably reversed in hesperetin (40 μmol/L) pretreatment (57.3% decreases in ATF4, 48.7% decreases in CHOP; Figure 4D,E). The AGE-elevated higher levels of ATF4 and CHOP decreased to 49.2 and 45.7%, respectively, when the cells were pretreated with 10 μmol/L rosiglitazone (Figure 4D,E).

### 3.5. Hesperetin Relieves the ER Stress-Mediated Apoptosis under AGE Stimulation

AGEs lower about 31.2% of Bcl-2 levels in SH-SY5Y cells (Figure 5A). Hesperetin (40 μmol/L) or rosiglitazone (10 μmol/L) upregulated the Bcl-2 level to 2.3-fold and 2.5-fold, respectively, in AGE-cultured SH-SY5Y cells (Figure 5A).

On the other hand, AGE-stimulated SH-SY5Y cells showed a 4.2-fold higher level of Bax protein in the cells. Hesperetin (40 μmol/L) or rosiglitazone (10 μmol/L) lowered the AGE-induced higher Bax levels in SH-SY5Y cells (48.2 and 52.7% reduction, respectively) (Figure 5A).

AGEs markedly decreased the Bcl-2/Bax ratio in SH-SY5Y cells, which was increased (4.1 and 4.6-fold elevation, respectively) by pretreatment with hesperetin (40 μmol/L) or rosiglitazone (10 μmol/L) (Figure 5A).

AGEs induced a 3.7-fold increase in caspase-12-like activity, which suffered a 2.1-fold and 1.9-fold reduction in SH-SY5Y cells pretreated with hesperetin (40 μmol/L) or rosiglitazone (10 μmol/L) (Figure 5B). The caspase-like activities in caspase-9 and caspase-3 increased by 4.1- and 4.0-fold in cells under AGE stimulation (Figure 5B). Hesperetin (40 μmol/L) downregulated the increased caspase-like activities of caspase-9 and caspase-3 exposed to AGEs to 56.7 and 47.1%, respectively (Figure 5B). The higher caspase-like activities of caspase-9 and caspase-3 caused by AGEs were lower to 52.9 and 41.8%, respectively, in the rosiglitazone (10 μmol/L) pretreated group (Figure 5B).

An elevation of apoptotic DNA fragmentation to 4.3-fold higher was induced by AGEs, which was decreased by 50.3% when pretreated with hesperetin (40 μmol/L) (Figure 5C). The pretreatment of rosiglitazone (10 μmol/L) alleviated the greater degree of apoptotic DNA fragmentation induced by AGEs (Figure 5C).

## 4. Discussion

AGE-induced oxidative stress significantly influences developing neurodegenerative disorders; thus, enhancing the antioxidant system to reduce oxidative stress could be a logical therapeutic approach [7]. Flavonoid compounds are clarified to antioxidative stress by their antioxidant activity, thus representing beneficial candidates for the protection against oxidative diseases, including neurodegenerative disorders, such as AD [32]. As an in vitro model for AD studies, the retinoic acid-induced cholinergic differentiation in SH-SY5Y cells was well used [24]. Recently, it has been demonstrated that G protein-coupled estrogen receptors have a direct anti-inflammatory effect in human cholinergic neurons [33]. Although flavonoids with estrogen-like activities could reduce inflammation in cholinergic neurons to consider as prospective neuroprotectants [34], the protective mechanisms related to alleviating oxidative stress and Aβ-mediated neurotoxicity induced by AGEs is essential to be elucidated. The obtained results show that hesperetin upregulates the antioxidative capacity to contribute to SH-SY5Y cells against ROS induced by AGEs. Further research on the ability to counter oxidative stress or examine the mechanisms by which hesperetin protects against AGE-induced neuronal cellular damage is valuable.

One characteristic hallmark of AD is an Aβ accumulation caused by the formation of plaque and Aβ complex toxicity [2,3]. Evidence has shown that AGE-induced ROS increases the synthesis of Aβ through the proteolytic processing of a transmembrane protein, APP, by upregulating APP processing proteins, such as BACE1, eventually leading to the aggregation and deposition of Aβ to cause neuronal damage [7]. The Aβ levels are highly dependent on Aβ degradation, although much attention has focused on the cellular production of Aβ by amyloidogenic processing. Among the enzymes that contributed to the cleaving of Aβ, NEP and IDE are the most important, which belong to two metalloprotease family members, M16 (IDE) and M13 (NEP), respectively [35]. Aβ composed a length of 39–43 amino acid residues; 40 amino acid residues (Aβ1-40), as well as 42 amino acid residues (Aβ1-42), are the predominant isoforms [36]. Remarkably, there is the potent aggregation and associated toxicity in Aβ1–42 [37]. In AGE-stimulated SH-SY5Y cells, hesperetin pretreatment resulted in the translational downregulation of APP and BACE1 accompanied by a decrease in Aβ secretion. It revealed that hesperetin might directly ameliorate Aβ overproduction via inhibiting Aβ generation. As mentioned before, the extent of Aβ expressed in the brain is controlled by Aβ production and its enzymatic degradation clearance pathways, thus investigating the influence of hesperetin on Aβ degradation. The pretreatment of AGE-cultured SH-SY5Y cells with hesperetin shown in the upregulation of protein levels in IDE and NEP, highlights that the attenuation of Aβ accumulation by this compound is mediated by modulating Aβ-degrading enzymes to increase Aβ degradation. Therefore, these findings support the notion that the downregulation of Aβ generation and promotion of Aβ degradation are responsible mechanisms by which hesperetin protects against AGE-induced neuronal cellular damage.

Evidence indicates that the accumulation of AGEs is associated with ER stress induction in cells contributing to the pathogenesis of AD and other multiple diseases [38]. During ER stress, GRP78 cooperated with the misfolded proteins to maintain the proteins in foldable states and control the release of three UPR mediators, including inositol-requiring enzyme 1 alpha (IRE1α), PERK, and ATF6 as well [39]. The CHOP protein is the transcription of genes controlled in cell apoptosis, activated by the PERK/eIF2α/ATF4 axis under prolonged ER stress [10]. Accordingly, several evidence lines suggest that AD’s pathogenesis and progression are closely linked with the UPR signaling pathway controlled byPERK [40]. Therefore, therapeutic targeting on the PERK-associated UPR signaling branches may help to develop a therapeutic approach to treat AD [40]. We observed that hesperetin pretreatment counters the AGE-induced upregulation of proteins under ER stress involving GRP78, p-PERK, p-eIF2α, and ATF4. Moreover, the increasing CHOP expression level in SH-SY5Y cells after the induction with AGEs was attenuated by hesperetin pretreatment. These data suggest the survivability of AGE-treated SH-SY5Y cells enhanced by hesperetin through suppressing ER stress-induced GRP78 to downregulate PERK/eIF2α/ATF4/CHOP, finally contributing to neuron protection.

The Bcl2-family protein expression is controlled by several transcriptional factors involving CHOP [41]. Although Bcl-2 family proteins are characterized on the mitochondrion, the proteins are localized at ER and nuclear membranes [42]. Hesperetin prevented protein downregulation in Bcl-2 and, in parallel, reduced Bax expression in SH-SY5Y cells under AGE stimulation. Therefore, the action of hesperetin on the reductions in AGE-induced ER stress may be associated with recovering the balance between the pro-and anti-apoptotic Bcl-2 family, which finally leads to inhibiting neuronal apoptosis. Prolonged ER stress can activate caspase-12 that may feed back in a caspase-9-dependent loop involving caspase-3 activation and certainly catalyze apoptosis [11]. We observed that AGEs significantly induced SH-SY5Y cell apoptosis and increased active caspase-12 that directly activated caspase-9 and the downstream effector caspase-3, supporting the ER stress-mediated activation of the caspase pathway involved in AGE-induced apoptosis [43]. Compared with AGE-treated SH-SY5Y cells, hesperetin-pretreated cells had a minor apoptotic DNA fragmentation and decreased along with the cleavage of caspase-12; in parallel, protein expression and the activity of cleaved caspase-9 and -3 expression were downregulated by hesperetin pretreatment. Our data suggest that hesperetin prevents the AGE-induced activation of apoptosis in SH-SY5Y cells by reducing the caspase-12-dependent ER stress.

There is a similar effect between hesperetin and positive rosiglitazone control against the Aβ aggravation of SH-SY5Y cell injury induced by AGE-mediated ER stress. In contrast to rosiglitazone playing a minor role in the regulation of Aβ generation [44], hesperetin lowered BACE1 protein expression to downregulate Aβ generation in SH-SY5Y cells induced by AGEs. Although PPARs have recently been identified as ahead of the times targets in researching new active drugs for AD [45], whether the activation of PPARs was involved in the neuron protection of hesperetin remains to be investigated further.

In conclusion, hesperetin protects SH-SY5Y cells against AGE-induced ROS, resulting in a decrease in the Aβ accumulation via the downregulation of Aβ generation and enhancing Aβ degradation. Additionally, hesperetin resolved PERK/eIF2α/ATF4/CHOP activation together with anti-apoptotic effects via the suppression of the caspase-12-dependent pathway in AGE-treated SH-SY5Y cells. Therefore, we conclude that hesperetin protects neuroblastoma cells against AGE-induced damage via multiple mechanisms, and thus may be an excellent option for potential pre-symptom treatment for neurodegenerative disorders augmented under glycation conditions.

## Figures and Tables

**Figure 1 nutrients-14-00745-f001:**
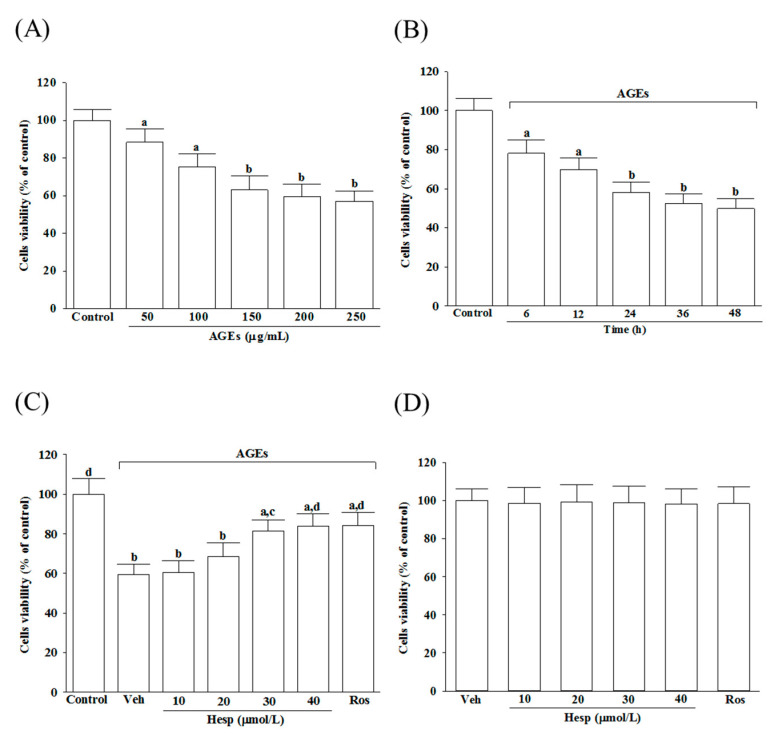
Changes to the reduced cell viability in AGE-induced SH-SY5Y cells. (**A**) SH-SY5Y cells were received 24 h prior to AGE exposure at various concentrations. (**B**) Incubation of SH-SY5Y cells with 200 μg/mL of AGEs for various periods. (**C**) Treatment of SH-SY5Y cells for 1 h with indicated concentrations of hesperetin (hesp; 10–40 μmol/L) or rosiglitazone (Ros; 10 μmol/L); after that, exposure to 200 μg/mL of AGEs for another 24 h. (**D**) Incubation of SH-SY5Y cells with indicated concentrations of hesp (10-40 μmol/L) or Ros (10 μmol/L) for 24 h without 200 μg/mL of AGE exposure. An MTT assay used cell viability determination; the value was expressed as a percentage of the control group from untreated cells. The results are presented as the mean ± SD of five independent experiments performed triplicate (n = 5). ^a^
*p* < 0.05 and ^b^
*p* < 0.01 when compared to the data from the untreated control group (control). ^c^
*p* < 0.05 and ^d^
*p* < 0.01 when compared to the data from the vehicle (Veh)-treated cells.

**Figure 2 nutrients-14-00745-f002:**
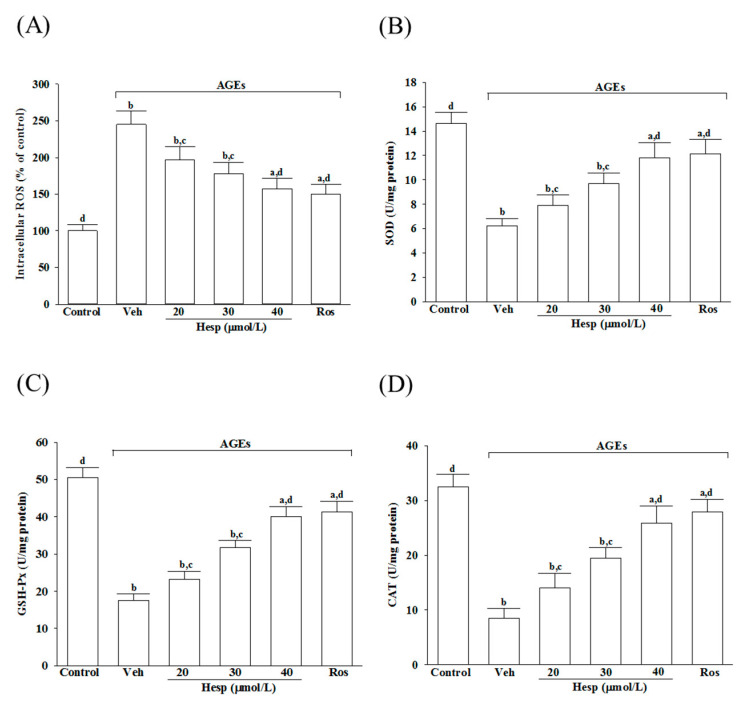
Influences on the ROS level and the antioxidant enzymes activities in SH-SY5Y cells under AGE stimulation. Cells were received 1 h pretreatment with hesperetin (hesp; 20–40 μmol/L) or rosiglitazone (Ros; 10 μmol/L), continued to 24 h AGE (200 μg/mL) exposure. (**A**) Intracellular ROS level was measured by the oxidation-sensitive fluoroprobe DCFH-DA. The activities of (**B**) SOD, (**C**) GSH-Px, and (**D**) CAT were qualified by commercial assay kits. The results are presented as the mean ± SD of five independent experiments performed triplicate (n = 5). ^a^
*p* < 0.05 and *^b^ p* < 0.01 when compared to the data from the untreated control group (control). ^c^
*p* < 0.05 and ^d^
*p* < 0.01 when compared to the data from the vehicle (Veh)-treated cells.

**Figure 3 nutrients-14-00745-f003:**
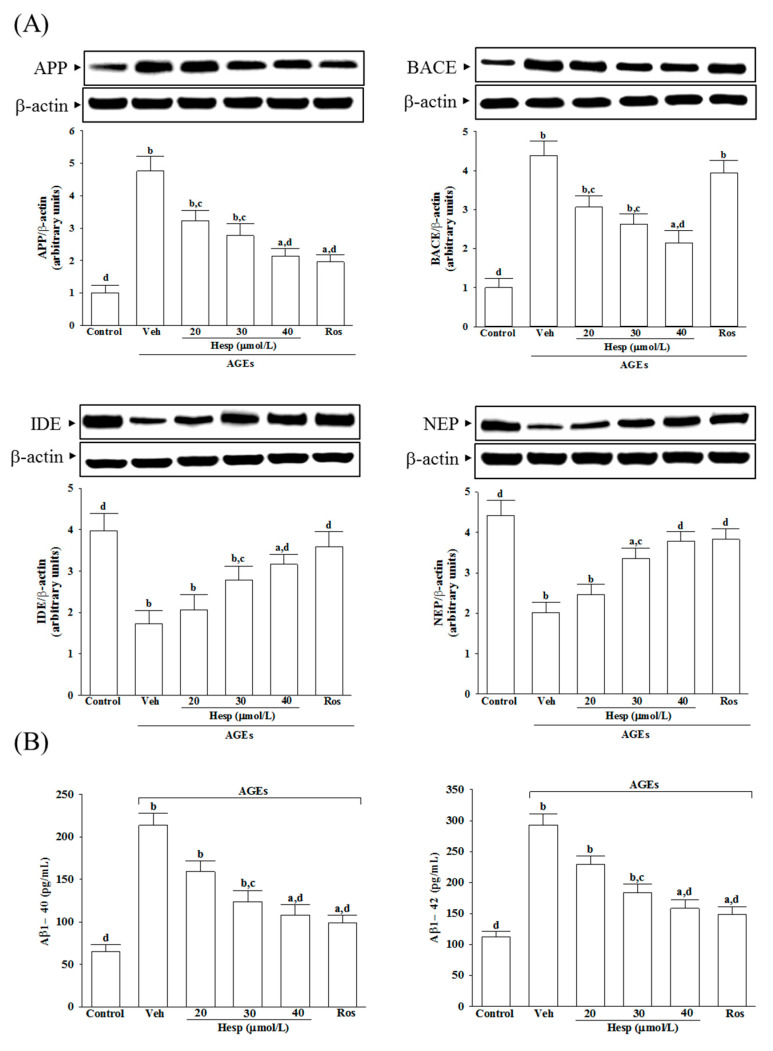
Changes in APP processing and Aβ secretion in SH-SY5Y cells under AGE stimulation. Cells were received 1 h pretreatment with hesperetin (hesp; 20–40 μmol/L) or rosiglitazone (Ros; 10 μmol/L), continued to 24 h AGE (200 μg/mL) exposure. (**A**) A representative Western blot analyzes the relative levels of amyloid precursor protein (APP), β-site APP-cleaving enzyme (BACE), insulin degrading enzyme (IDE), and neprilysin (NEP). Band densities were normalized with β-actin. (**B**) ELISA assay was used to determine the level changes in Aβ1-40 and Aβ1-42. The results are presented as the mean ± SD of five independent experiments performed triplicate (n = 5). ^a^
*p* < 0.05 and ^b^
*p* < 0.01 when compared to the data from the untreated control group (control). ^c^
*p* < 0.05 and ^d^
*p* < 0.01 when compared to the data from the vehicle (Veh)-treated cells.

**Figure 4 nutrients-14-00745-f004:**
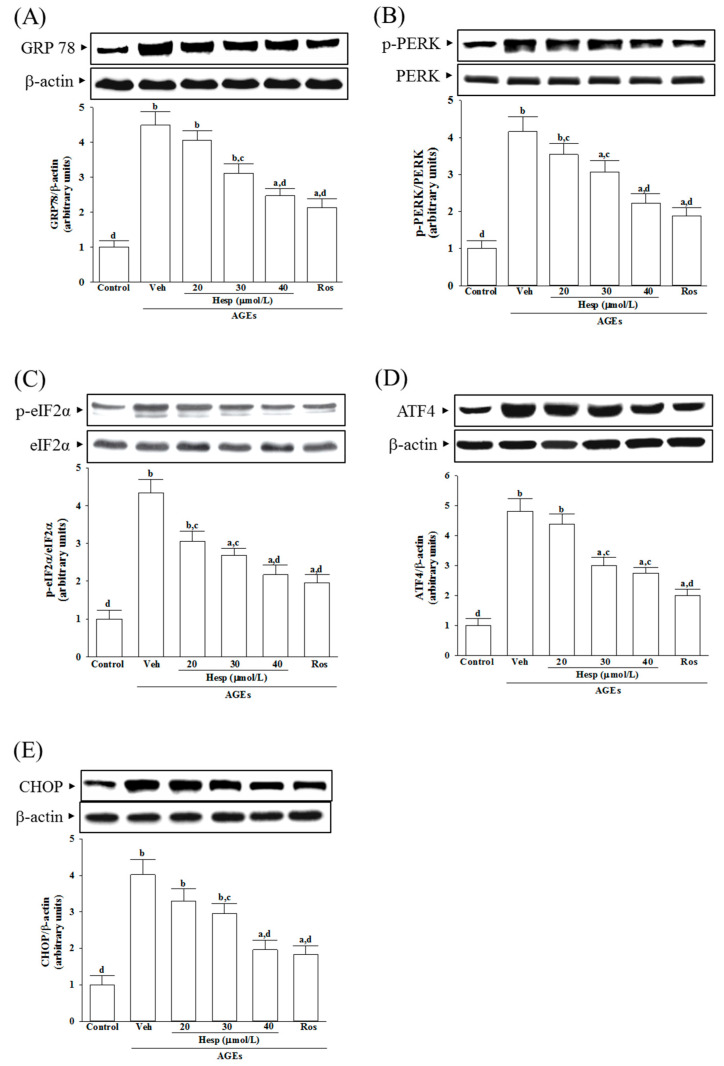
Investigation on the related molecules in ER stress in SH-SY5Y cells under AGE stimulation. Cells were received 1 h pretreatment with hesperetin (hesp; 20–40 μmol/L) or rosiglitazone (Ros; 10 μmol/L), continued to 24 h AGE (200 μg/mL) exposure. A representative Western blot analyzed the relative levels of (**A**) 78-kDa glucose-regulated protein(GRP78), (**B**) p-protein kinase-like endoplasmic reticulum kinase(PERK), PERK, (**C**) p-eukaryotic initiation factor (eIF) 2α, eIF2α, (**D**) activating transcription factor (ATF)4, and (**E**) C/EBP homologous protein (CHOP). The band densities for protein were normalized to the β-actin band intensity. The ratio of phosphoprotein to total protein in PERK (p-PERK/PERK) and eIF2α (p-eIF2α/eIF2α) is also shown. The results are presented as the mean ± SD of five independent experiments performed triplicate (n = 5). ^a^
*p* < 0.05 and ^b^
*p* < 0.01 when compared to the data from the untreated control group (control). ^c^
*p* < 0.05 and ^d^
*p* < 0.01 when compared to the data from the vehicle (Veh)-treated cells.

**Figure 5 nutrients-14-00745-f005:**
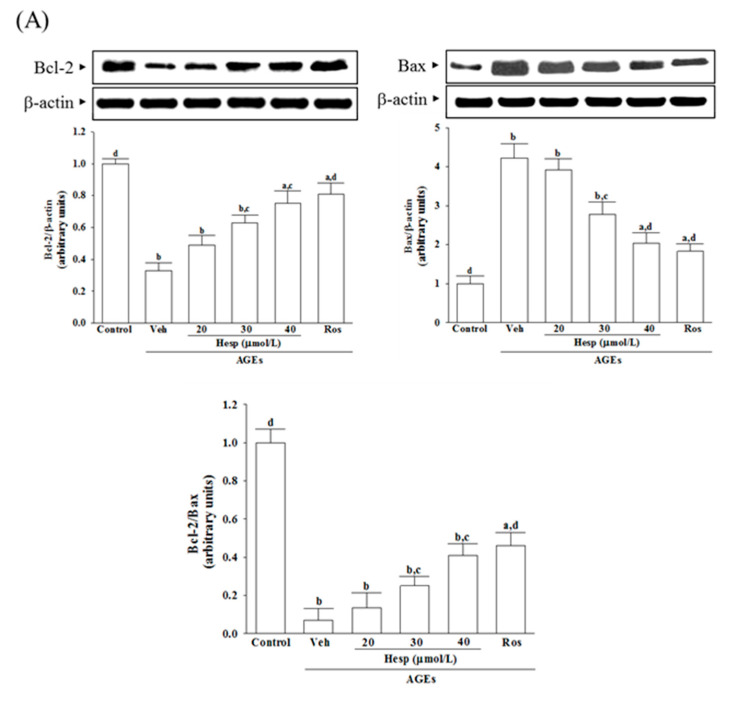
Analysis of ER stress-associated apoptosis in SH-SY5Y cells induced by AGEs. Cells were received 1 h pretreatment with hesperetin (hesp; 20–40 μmol/L) or rosiglitazone (Ros; 10 μmol/L), continued to 24 h AGE (200 μg/mL) exposure. (**A**) A representative Western blot analyzes the relative levels of Bcl and Bax. Band densities were normalized with β-actin. The relative intensities of Bcl-2 to Bax (Bcl-2/Bax) were shown. (**B**) Analysis of caspase-12, -9, and -3-like activity. The caspase-12-like activity in cell lysates was quantified by fluorescent detection of the cleavage of substrate ATAD-AFC. The activities in caspase-9 and caspase-3 were evaluated by the cleavage of their respective substrates, Ac-LEHD-pNA and Ac-DEVD-pNA. (**C**) Quantification of apoptotic DNA fragmentation. The commercial kit quantified cytosolic histone-associated DNA fragmentation. The results are presented as the mean ± SD of five independent experiments performed triplicate (n = 5). ^a^
*p* < 0.05 and ^b^
*p* < 0.01 when compared to the data from the untreated control group (control). ^c^
*p* < 0.05 and ^d^
*p* < 0.01 when compared to the data from the vehicle (Veh)-treated cells.

## Data Availability

All the data needed to evaluate the conclusions in the paper are present in the paper. Additional data related to this paper may be requested from the authors.

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
