# Peer review of "The Citrus Flavonoid Hesperetin Encounters Diabetes-Mediated Alzheimer-Type Neuropathologic Changes through Relieving Advanced Glycation End-Products Inducing Endoplasmic Reticulum Stress"

_nutrients, 2022, doi:10.3390/nu14040745_

Round 1

Reviewer 1 Report

Dear Authors, 

The research paper submitted is very interesting highlighting the new and emerging role of flavonoids in Alzheimer disease. However, the present study needs a revision in order to improve it.

Here are my concerns:

1- Introduction section: It should be increased, better highlighting the role of general flavonoids in different field of application, such as protecting against development of dementia (10.1023/a:1007614613771), promoting the cell survival in stress condition (nutrient deprivation and hypoxia, which characterize the neurodegenerative processes) of neural stem cells (10.1016/j.acthis.2018.01.003), and, of course, their emerging role as therapeutics against Alzheimer disease (10.1155/2018/7043213; 10.3233/BPL-200098; 10.1093/ajcn/nqaa079).

2- Furthermore, concerning again the introduction section, since the Authors rightly mentioned that flavonoids owns anti-inflammatory and anti-oxidative properties, they should quote literature concerning the pivotal role of oxidative stress and inflammation in the onset/progression of  Alzheimer disease (10.1021/bi000169p; 10.1155/S1110724302203010; 10.1021/acschemneuro.9b00333; 10.3390/ijms21176128).

3- The Authors used the SH-SY5Y cell line for their experiments after retinoic acid differentiation. This is fine, especially because, as reported in literature, the SH-SY5Y are widely used for neurobiology study and retinoic acid is the most common used treatment to induce differentiation (10.1007/978-1-62703-640-5_2). Moreover, retinoic acid differentiation induce the cholinergic phenotype of this cell line (10.1023/a:1022848718109; 10.1007/s12035-019-1605-3). On this regard, since the flavonoids have estrogen-like activities (10.2174/0929867043365251), and recently it has been demonstrated that G protein-coupled estrogen receptor (GPER1) have a direct anti-inflammatory effect of in human cholinergic neurones (10.1111/jne.12837), the Authors should mention this estrogen-like acitivity exerted by flavonoids at least in the discussion section in order to better underline and justify the anti-inflammatory effect of flavonoid in cholinergic neurons. 

4- General comments on figures: Sometimes the Authors use a grey square edge for the histograms (such as in figures 3 and 4). Please, remove it since in order to avoid the overlay with the text or blots. 

5- General comments on figure legends: Please, when different panels are quoted (such as A, B, C, D and E), use the same font (bold or regular).

6- Figure 2 legend (page 7, line 265): edit "D)" with "(D)".

7- Figure 5: the panel A (concerning the BAX and Bcl-2 western blotting analysis) is not present in the current version of the manuscript. Please, provide it.

8- General comments on the main manuscript: Please, avoid any typos that are present, such as different font (e.g.: page 2, line 67 (AGE/RAGE signaling); page 3, line 138 (fluorescence intensity); and many others) and font size (e.g.: page 3, line 120 (Cell Viability Assay); page 4, line 153 (absorbance); and many others).

Please, check carefully all the typos and fix it.

7-

Author Response

Reply to Reviewer 1

Ms. Ref. No.:  nutrients-1581987 R1
Title: The Citrus Flavonoids Hesperetin Encounters Diabetes-Mediated Alzheimer-Type Neuropathologic Changes through Relieving Advanced Glycation End-Products Induce Endoplasmic Reticulum Stress

Authors: Lai MC et al.

Dear distinguished referee:

Thank you very much for reading this manuscript and the helpful comments. The revision has been amended according to your kind suggestions as follows:

1. Introduction section: It should be increased, better highlighting the role of general flavonoids in different field of application, such as protecting against development of dementia (10.1023/a:1007614613771), promoting the cell survival in stress condition (nutrient deprivation and hypoxia, which characterize the neurodegenerative processes) of neural stem cells (10.1016/j.acthis.2018.01.003), and, of course, their emerging role as therapeutics against Alzheimer disease (10.1155/2018/7043213; 10.3233/BPL-200098; 10.1093/ajcn/nqaa079).

Reply

The role of flavonoids in different fields of application has been highlighted according to your kind instructions (line 79-82). We hope this improvement will be satisfactory.

2. Furthermore, concerning again the introduction section, since the Authors rightly mentioned that flavonoids owns anti-inflammatory and anti-oxidative properties, they should quote literature concerning the pivotal role of oxidative stress and inflammation in the onset/progression of Alzheimer disease (10.1021/bi000169p; 10.1155/S1110724302203010; 10.1021/acschemneuro.9b00333; 10.3390/ijms21176128).

Reply

According to your instructions, the indicated section (lines 86-89, 92-95) has been revised. We hope you consider this change sufficient and acceptable.

3. The Authors used the SH-SY5Y cell line for their experiments after retinoic acid differentiation. This is fine, especially because, as reported in literature, the SH-SY5Y are widely used for neurobiology study and retinoic acid is the most common used treatment to induce differentiation (10.1007/978-1-62703-640-5_2). Moreover, retinoic acid differentiation induce the cholinergic phenotype of this cell line (10.1023/a:1022848718109; 10.1007/s12035-019-1605-3). On this regard, since the flavonoids have estrogen-like activities (10.2174/0929867043365251), and recently it has been demonstrated that G protein-coupled estrogen receptor (GPER1) have a direct anti-inflammatory effect of in human cholinergic neurones (10.1111/jne.12837), the Authors should mention this estrogen-like acitivity exerted by flavonoids at least in the discussion section in order to better underline and justify the anti-inflammatory effect of flavonoid in cholinergic neurons. 

Reply

The role of estrogen-like activity on the anti-inflammatory effect of flavonoids in cholinergic neurons has been mentioned in the Discussion section following your instructions. Please find it on lines 414-419. We hope you consider this improvement sufficient and acceptable.

4. General comments on figures: Sometimes the Authors use a grey square edge for the histograms (such as in figures 3 and 4). Please, remove it since in order to avoid the overlay with the text or blots. 

Reply

The grey square edge for the histograms (figures 3, 4, 5) has been removed following your instructions. We honestly wish it would be suitable to meet your requirements.

5. General comments on figure legends: Please, when different panels are quoted (such as A, B, C, D and E), use the same font (bold or regular).

Reply

The writing format on figure legends has been unified. Thank for the instructions.

6. Figure 2 legend (page 7, line 265): edit "D)" with "(D)".

Reply

The indicated missing has been corrected (line 281). Thanks very much.

7. Figure 5: the panel A (concerning the BAX and Bcl-2 western blotting analysis) is not present in the current version of the manuscript. Please, provide it.

Reply

Thanks for reminding me, the panel A in Figure 5 has been presented.

8. General comments on the main manuscript: Please, avoid any typos that are present, such as different font (e.g.: page 2, line 67 (AGE/RAGE signaling); page 3, line 138 (fluorescence intensity); and many others) and font size (e.g.: page 3, line 120 (Cell Viability Assay); page 4, line 153 (absorbance); and many others). Please, check carefully all the typos and fix it.

Reply

The writing format throughout the whole manuscript has been unified. Thank for the instructions.

The changes in the revision are highlighted in red. We hope that this revised version of our work will meet your high standards for acceptance. Also, I wish to express my warmest thanks to you again. Your kind agreement of recognition will be sincerely appreciated.

Reviewer 2 Report

The paper is well written, contains 5 figures and all the sections.  Its title, The Citrus Flavonoids Hesperetin Encounters Diabetes-Medi-2 ated Alzheimer-Type Neuropathologic Changes through Re-3 living Advanced Glycation End-Products Induce Endoplasmic 4 Reticulum Stress 

no major spelling mistakes were detected it describes clearance of a beta and oxidative stress, the conclusions - 

 The Citrus Flavonoids Hesperetin Encounters Diabetes-Medi-2 ated Alzheimer-Type Neuropathologic Changes through Re-3 lieving Advanced Glycation End-Products Induce Endoplasmic 4 Reticulum Stress  + 

Flavonoid compounds with antioxidant activity are 402 known to reduce oxidative stress, thus representing beneficial candidates for protection 403 against oxidative diseases.

4 minor comments,

1.The authors can add a citation to the following paper: 

The amyloid hypothesis of Alzheimer's disease at 25 years. Selkoe DJ, Hardy J.EMBO Mol Med. 2016 Jun 1;8(6):595-608. doi: 10.15252/emmm.201606210. Print 2016 Jun.PMID: 27025652  2. perhaps color the bars in the figures for clearer presentation using different colors  3 add funding sources 4 add that the data will be availabe according to a reasonable request 

Author Response

Reply to Reviewer 2

Ms. Ref. No.:  nutrients-1581987 R1
Title: The Citrus Flavonoids Hesperetin Encounters Diabetes-Mediated Alzheimer-Type Neuropathologic Changes through Relieving Advanced Glycation End-Products Induce Endoplasmic Reticulum Stress

Authors: Lai MC et al.

Dear distinguished referee:

Thank you very much for reading this manuscript and the helpful comments. The revision has been amended according to your kind suggestions as follows

1. The authors can add a citation to the following paper: 

The amyloid hypothesis of Alzheimer's disease at 25 years. Selkoe DJ, Hardy J.EMBO Mol Med. 2016 Jun 1;8(6):595-608. doi: 10.15252/emmm.201606210. Print 2016 Jun.PMID: 27025652 

Reply

The indicated reference has been cited in ref. 3 (line 519) according to your instructions. Thank you very much.

2. perhaps color the bars in the figures for clearer presentation using different colors 

Reply

A clear description mark has been presented under each bar in the figures. We hope this will be satisfactory for the requirements.

3. add funding sources

Reply

Funding sources have been shown in lines 510-511. We hope this will be satisfactory for the requirements.

4. add that the data will be availabe according to a reasonable request 

Reply

Data availability statement has been presented in lines 511-512. We hope this will be satisfactory for the requirements.

The changes in the revision are highlighted in red. We hope that this revised version of our work will meet your high standards for acceptance. Also, I wish to express my warmest thanks to you again. Your kind agreement of recognition will be sincerely appreciated.